# Enhanced Risk of Osteoporotic Fracture in Patients with Sarcopenia: A National Population-Based Study in Taiwan

**DOI:** 10.3390/jpm12050791

**Published:** 2022-05-13

**Authors:** Han-Wei Zhang, Zhi-Ren Tsai, Ko-Ta Chen, Sheng-Lun Hsu, Yi-Jie Kuo, Ying-Chin Lin, Shu-Wei Huang, Yu-Pin Chen, Hsiao-Ching Peng, Jeffrey J. P. Tsai, Chun-Yi Hsu

**Affiliations:** 1Biomedica Corporation, New Taipei 23146, Taiwan; omnizhang@outlook.com (H.-W.Z.); jainepeng@biomdcare.com (H.-C.P.); 2Ph.D. Program for Aging, China Medical University, Taichung 40402, Taiwan; 3Institute of Population Health Sciences, National Health Research Institutes, Miaoli 35053, Taiwan; 4Department of Electrical and Computer Engineering, Institute of Electrical Control Engineering, National Yang Ming Chiao Tung University, Hsinchu 30010, Taiwan; 5Department of Computer Science & Information Engineering, Asia University, Taichung 41354, Taiwan; ren@asia.edu.tw; 6Department of Medical Research, China Medical University Hospital, China Medical University, Taichung 40402, Taiwan; 7Center for Precision Medicine Research, Asia University, Taichung 41354, Taiwan; jjptsai@gmail.com; 8Department of Orthopedics, Taipei Medical University Hospital, Taipei 11031, Taiwan; kotahero@gmail.com; 9Department of Orthopedics, School of Medicine, College of Medicine, Taipei Medical University, Taipei 11031, Taiwan; benkuo5@gmail.com; 10Department of Family Medicine, Wan Fang Hospital, Taipei Medical University, Taipei 11696, Taiwan; 106182@w.tmu.edu.tw (S.-L.H.); 89055@w.tmu.edu.tw (Y.-C.L.); 11Department of Orthopedics, Wan Fang Hospital, Taipei Medical University, Taipei 11696, Taiwan; judyya1022@gmail.com; 12Department of Family Medicine, School of Medicine, College of Medicine, Taipei Medical University, Taipei 11031, Taiwan; 13Department of Geriatric Medicine, School of Medicine, College of Medicine, Taipei Medical University, Taipei 11031, Taiwan; 14Department of Bioinformatics and Medical Engineering, Asia University, Taichung 41354, Taiwan; 15Graduate Institute of Biomedical Science, China Medical University, Taichung 40402, Taiwan; hsuc@mail.cmu.edu.tw

**Keywords:** sarcopenia, fracture, osteoporosis, incidence, nationwide population-based study

## Abstract

Sarcopenia is a progressive and generalized skeletal muscle disorder associated with poor health outcomes in older adults. However, its association with the risk of fracture risk is yet to be clarified. Therefore, this study aimed to assess the incidence and consequence of osteoporosis-related fractures among patients with sarcopenia in Taiwan. A retrospective, population-based study on 616 patients with sarcopenia, aged >40 years, and 1232 individuals without sarcopenia was conducted to evaluate claims data from Taiwan’s National Health Insurance Research Database collected in the period January 2000–December 2013. The incidence rate of osteoporosis-related fracture was 18.13 and 14.61 per 1000 person years in the patients with sarcopenia and comparison cohort, respectively. Patients with sarcopenia had a greater osteoporotic fracture risk (adjusted hazard ratio [HR] 2.11; 95% confidence interval [CI] 1.47–3.04) after correcting for possible confounding. Additionally, females showed statistically significant correlations of sarcopenia with osteoporosis-related fracture risk (HR 1.53; CI 0.83–2.8 for males and HR 2.40, CI 1.51–3.81 for females). During this retrospective study on the fracture risk in Taiwan, an adverse impact of sarcopenia was observed, which substantiates the need to work toward sarcopenia prevention and interventions to reverse fracture susceptibility in patients with sarcopenia.

## 1. Introduction

Sarcopenia is a syndrome illustrated by the loss of skeletal muscle mass and function with increasing age or secondary to disease [1]. Clinicians are familiar with sarcopenia as a well-known predictor of all-cause mortality among community-dwelling older people [2]. Sarcopenia is also associated with poor health outcomes, including disability, morbidity, and impaired quality of life [3] and may result in unfavorable postoperative prognosis, including increased complication rates, mortality, and morbidity in major surgical procedures [4,5]. The potential adverse effects and outcomes highlight the importance of sarcopenia in the health care of older people.

Older people with sarcopenia are also more susceptible to falls [6] because of reduced muscle strength, which makes it harder to regain lost balance [7]. Sarcopenia can decrease the mechanical loading of the skeleton and lead to reduced adaptive bone remodeling [8]. Therefore, sarcopenia is closely linked to osteoporosis, and their combined effect may exacerbate negative health outcomes [9]. In fact, sarcopenia has also been identified as a predictor of fracture risk [10]. However, most of the recent studies on the association of sarcopenia with fracture risk were cross-sectional or small-scale study designs [10]. Evidence from large-scale studies with longitudinal follow-up is scarce and is needed to verify the results of the smaller studies.

The present study investigated the correlation between sarcopenia and fracture risk using national population-based data of patients in Taiwan. The cohort study design focused on patients with sarcopenia aged older than 40 years, to determine the incidence rate and relative risk of developing osteoporosis-related fractures compared with the rate and risk in healthy individuals over a 13-year follow-up.

## 2. Materials and Methods

### 2.1. Data Source

The data for one million randomly tested recipients from the period 1996–2013 were included in this study and were acquired from the Longitudinal Health Insurance Database (LHID) within Taiwan’s National Health Insurance Research Database (NHIRD). The observation period was between 2000 and 2013 to improve the consistency of the NHIRD data. This study was approved by the Institutional Review Board Ethics Committee at the China Medical University (approval # CMUH104-REC2-115 (CR-6)) and the written informed consent was waived due to the use of de-identified/anonymized data from the LHID database.

### 2.2. Study Design and Population

One million outpatient and inpatient claim data obtained from the LHID were included in this study. Among them, patients with missing/unidentified sex and birth month/year records were excluded. The remaining 857,097 individuals who had claims between 1 January 2000 and 31 December 2013 were enrolled in this study. A total of 560,251 patients were excluded for the following reasons: (1) people aged <40 years at the beginning of the follow-up period; (2) patients with osteoporotic fractures that occurred before the beginning of the follow-up period; (3) patients who had been diagnosed with pathological fractures, Paget’s disease, malignancy, and accidents before the start of follow-up; (4) individuals without medical claim records within the follow-up period. Additionally, this study excluded those who had medical claims (less or equal to two outpatient visits) of sarcopenia in the comparison cohort to ensure individuals unexposed in sarcopenia. Ultimately, this study included 295,637 individuals. Figure 1 shows the flow chart for the selection of the study population.

### 2.3. Selection and Length of Exposure

Taiwan’s NHIRD was established in 1996 and contains all medical records and insurance claims data of 99% of the Taiwanese population covered by the universal health insurance program [11]. The NHIRD holds data on the clinical outcomes of real-world practice and is distinguished for its implication and clinical impact beyond the results obtained from clinical trials [12]. The National Health Research Institute has established firm regulations and levy penalties to avert any medical fraud in clinical practice. Subsequently, NHIRD delivers high-quality reliable health care information for investigating real-world indications based on a substantial amount of wellbeing data analytics. The database registers the health status of all individuals according to the International Classification of Disease, 9th Revision, Clinical Modification (ICD-9-CM).

Individuals who were newly diagnosed with sarcopenia (ICD-9-CM code: 728.2) between 2000 and 2013 formed the sarcopenia cohort by reporting at least three outpatient visits or one hospitalization for sarcopenia. The comparison cohort comprised individuals without sarcopenia. The cohort was formed using precise pairing at a ratio of 1:2 according to the birth year and month, sex, and index year. A total of 1848 individuals were appointed for the two cohorts, with 616 and 1232 individuals included in the sarcopenia and control cohorts, respectively. For all subsequent analyses, the very first date of visit to the outpatient clinic or hospitalization referring to a diagnosis of sarcopenia and the first claim date in the follow-up period were defined as the index date for the sarcopenia and control cohort, respectively. The study subjects were followed from the start of study in 2000 until 2013.

### 2.4. Definition of the Frequency of Osteoporotic Fracture

The osteoporosis-related fracture incidence rate was conveyed as the number of incidents per 1000 person years in both sarcopenia and comparison cohorts. The primary outcome measure was new cases of osteoporosis-related fracture diagnosed throughout the study period based on ICD-9-CM codes 733.8, 733.93, 733.96–733.98, 805–813, 819.0, 820–824, 905.1–905.4, 952.00, 952.05, 952.10, 952.13–952.15, 952.18, 952.19, 952.2, 952.3, 952.8, 952.9, and V54.8. Those patients were defined at least three separate medical claims in the outpatient clinic in order to minimize miscoding in one claim issued in an inpatient hospitalization or in the outpatient reimbursement data. The very first hospitalization or outpatient clinic visit date with osteoporotic fracture diagnosis was identified as the date of diagnosis and also considered as the date of recently diagnosed osteoporosis-related fracture for all successive analyses. The osteoporosis-related fracture diagnosis, death, or 31 December 2013, whichever occurred first, were used as the endpoint from the index date.

### 2.5. Potential Confounding Factors

Comorbidity was one of the confounding factors considered in the present study to lessen the impact of the data selection bias. Patients with comorbidities were those who were defined based on the diagnosis history acquired from a minimum of three outpatient visits or at least one hospitalization before the period of newly diagnosed osteoporotic fracture. Comorbidities considered in the analyses were as follows: rheumatoid arthritis (714.0), smoking habit (305.1, 491.0, 491.2, 492.8, 496, 523.6, 989.84, V15.82, and 649.0), alcohol use (265.2, 291, 303, 305, 357.5, 425.5, 535.3, 571.0–571.3, 980.0, V11.3, 790.3, and A215), hypertension (401–405, and 437.2), diabetes (249, 250, 648.8, 648.0, and A181), dyslipidemia (272), obesity (278, 646.1, 649.1, 649.2, V45.86, V65.3, and V77.8), coronary heart disease (410–414), stroke (362.34, 430–438, procedure 38.12, 38.42), chronic obstructive pulmonary disease (490–496), depression (296.2, 296.3, 296.5, 296.82, 300.4, 309, and 311), cognitive dysfunction (290, 294.1, 294.8, 294.9, 310.1, and 331), and Parkinson’s disease (332). In addition, this study also considered the confounding effect of drug therapy (based on the Anatomical Therapeutic Chemical classification), which was defined as patients who have received at least one outpatient or one inpatient treatment prior to the newly diagnosed osteoporosis-related fracture. The drug treatment considered in the analysis was glucocorticoids (H02AB, R03BA). Adjustments for all confounding factors were performed in the analyses to avert potential bias due to these confounding factors.

### 2.6. Statistical Analysis

The descriptive data were shown with the mean ± standard deviation for continuous variables and as frequencies with percentages (%) for categorical variables. The differences in demographic characteristics and comorbidities between the sarcopenia cohorts and matched control were evaluated using the chi-square test and Student’s *t*-test. The osteoporosis-related fracture risk between the two cohorts was evaluated with the Cox proportional hazards regression model. This study anticipated the independent effects of sarcopenia on osteoporosis-related fracture incidents by adjusting for age, sex, urbanization level, comorbidities, and drug treatments. Furthermore, this study investigated whether the effects differed between male and female genders in a stratified analysis. Kaplan–Meier analysis was performed to elucidate the possibility of individuals developing an osteoporosis-related fracture during the follow-up period, and the log-rank test was employed to evaluate the difference between the two study cohorts. All analyses were achieved using the Meta Trial Platform (Version 1.0.0; Biomedica Corporation; New Taipei, Taiwan) and QCheck Solution (Version 3; Biomedica Corporation; New Taipei, Taiwan), and Statistical Product and Service Solutions software (SPSS; Version 22; IBM SPSS Statistics for Windows; IBM Corp., Armonk, NY, USA), and all *p* values ≤ 0.05 were considered statistically significant.

## 3. Results

### 3.1. Clinical Details of the Study Population

The demographic data and comorbid conditions of both cohorts were summarized in Table 1. The mean ages in the patients with sarcopenia (60.41 ± 12.00 years) and the comparison cohort (60.37 ± 12.00 years) were similar. The matched-patient analysis between the sarcopenia and control cohorts revealed significant differences for the parameters of urbanization level and many of the comorbidities including rheumatoid arthritis, hypertension, diabetes, dyslipidemia, obesity, coronary heart disease, stroke, chronic obstructive pulmonary disease, cognitive dysfunction and Parkinson’s disease. Expectedly, patients with sarcopenia generally had higher prevalence of comorbidities than those in the control cohort.

### 3.2. Associations between Sarcopenia and Osteoporosis-Related Fracture

The correlation of osteoporosis-related fracture risks between the two cohorts (patients with sarcopenia and control subjects without sarcopenia) was listed in Table 2. In the sarcopenia cohort, the incidence rate of osteoporosis-related fractures was higher than the comparison cohort (18.13 vs. 14.61 per 1000 person years, respectively). The crude HR for osteoporosis-related fractures (1.23; 95% CI, 0.90–1.67; *p* = 0.188) in the sarcopenia cohort was higher than that in the comparison cohort; in addition, after correcting for the possible confounding caused by demographic variables and comorbidities, the adjusted HR for osteoporosis-related fractures (2.11; 95% CI, 1.47–3.04; *p* < 0.001) was significantly greater in the sarcopenia cohort than in the comparison cohort, implying that, among individuals aged ≥40 years, the likelihood of being diagnosed with an osteoporosis-related fracture was 111% higher in patients with sarcopenia than in those without sarcopenia. Moreover, the adjusted HRs (95% CIs) for osteoporosis-related fracture in the sarcopenia cohorts were 1.53 (0.83–2.82; *p* = 0.174) for males and 2.40 (1.51–3.81; *p* < 0.001) for females. Following the data stratification by sex, sarcopenic female patients showed statistically significant correlations with osteoporosis-related fractures compared to those in the comparison cohorts.

The suitability of the Cox proportional hazards model is supported by the plot in Figure 2, in which the log [−log (survival function)] versus log of survival time is plotted for sarcopenia. During the 13-year follow-up, the osteoporosis-related fracture incidence rate, evaluated using the Kaplan–Meier method (Figure 3), was higher in the sarcopenia cohort than in the control cohort (log-rank test, *p* = 0.0187).

## 4. Discussion

During the 13-year follow-up, the risk of developing osteoporosis-related fracture was significantly greater in patients with sarcopenia, who were characterized by more underlying comorbidities, than in individuals without sarcopenia. In addition, in this population-based study, female patients with sarcopenia were more susceptible to osteoporosis-related fracture than male patients with sarcopenia.

With greater awareness of the negative effects of sarcopenia in geriatric populations, sarcopenia has attracted increasing attention from clinicians and recently received a specific International Classification of Disease, 10th Revision, Clinical Modification (ICD-10-CM) code to better distinguish sarcopenia from similar or coexisting diseases that manifest with muscular wasting [13]. However, the ICD-10-CM code has been generally accepted for use in Taiwan since 2020. In other words, there is no specific disease code for sarcopenia in our population-based data collected from Taiwan’s NHIRD, in which ICD-9-CM was the main disease coding system before 2020. However, this study utilized the ICD-9-CM code 728.2 (muscular wasting and disuse atrophy, not elsewhere classified) as the representative code, with strict selection criteria for the main diagnostic coding and number of outpatient visits or hospitalization; this was important in order to identify the potential patients with sarcopenia and minimize coding errors or interference from similar or coexisting diseases, acting in concert with the same methods used in the previously published population-based study [14,15]. The evidence from this study should have high reliability because of the large number of patients and long follow-up time to assess the influence of sarcopenia on fracture risk.

Sarcopenia is a musculoskeletal syndrome referred to increasing age or secondary to disease [1]. However, several chronic diseases may be also directly or indirectly associated with sarcopenia. Evidence has disclosed that high prevalence of sarcopenia is found in patients with several age-related diseases including cardiovascular diseases, dementia, diabetes and respiratory disorder [16]. Several mechanisms including decreased activity, sedentary behavior, inflammation and malnutrition might explain the correlation between sarcopenia and these age-related diseases. Decreased physical activity and sedentary life style, which are well-known risk factors for sarcopenia [17], are common in older population [18] and also highly prevalent in patients with chronic heart failure [19], diabetes [20] and respiratory diseases [21]. In addition, chronic inflammation is characteristic of aging process, deeply connected with age-related diseases and shown to prompt muscle wasting and pathological muscle loss [22]. On the other hand, older patients with cognitive disorders are susceptible to malnutrition [23], which is also a critical factor resulting in sarcopenia [24]. All the above-mentioned cofounding factors may therefore result in the sarcopenic patients concomitantly suffering more comorbidities, acting in concert with our finding that older patients with sarcopenia had higher prevalence of rheumatoid arthritis, hypertension, diabetes, dyslipidemia, obesity, coronary heart disease, stroke, chronic obstructive pulmonary disease, cognitive dysfunction and Parkinson’s disease than those without sarcopenia.

The consequence of high fracture risk in patients with sarcopenia can be explained by the poor muscle function and impaired standing balance resulting from the disease nature of sarcopenia [25] and co-occurrence of osteoporosis and sarcopenia resulting in poor bone density susceptible to osteoporosis-related fracture [26]. Falling accidents are a direct cause of fractures in the elderly. Impaired standing steadiness is a strong risk factor for falls in older people [27]. Muscle strength and muscle mass were positively linked with the ability to maintain vertical steadiness in older adults [28,29], which is consistent with the finding from a meta-analysis that sarcopenic individuals had a significantly higher risk of falls [30]. Conversely, osteoporosis and sarcopenia are interrelated in geriatric patients [31,32]. Both diseases share an underlying pathology and reinforce each other in terms of negative clinical outcomes [33,34]. The high co-occurrence of osteoporosis and sarcopenia in elderly people may predispose them to fall accidents, and thereby increase the risk of osteoporotic fractures.

Our study also showed that the fracture risk was higher in individuals of the female sex with sarcopenia than in individuals of the male sex. This sex difference may be explained by the higher prevalence of osteoporosis in older women than in older men with sarcopenia, which may be justified by the earlier estrogenic deprivation caused by menopause in females. In addition, the decline in estrogen levels associated with menopause could not only cause a rapid loss of bone density but also lead to a decrease in muscle mass and muscle strength [35], which echoes the finding by Frisoli et al. of a higher prevalence of osteoporosis in sarcopenic women than sarcopenic men [36]. However, a recent meta-analysis claimed a relatively higher fracture risk in males than in female patients with sarcopenia (odds ratio: 2.52 versus 1.98), albeit with high heterogeneity among the enrolled studies [30]. The inconsistency of the sex difference findings on the fracture susceptibility in patients with sarcopenia between our report and the literature may have resulted from differences in the patient selection, race percentages, and varied diagnostic criteria for sarcopenia, so further evidence is needed from more robust study designs on patient selection and follow-up to assess the impact of sex on the risk of fracture in patients with sarcopenia.

Only a few studies with longitudinal follow-ups have described the correlation of sarcopenia with fracture. In two longitudinal follow-up studies on community-dwelling older men, Scott et al. reported that the fracture risk over 6 years was higher in sarcopenic overweight men than in non-sarcopenic obese men in a cohort with 1486 Australian participants [37]. Yu et al. disclosed that sarcopenia was associated with an increased fracture risk independent of bone density and other clinical risk factors in a cohort of 2000 Hong Kong male participants over 65 years of age. In addition, a large cohort of 913 healthy 65-year-old community residents in Switzerland revealed that a low lean mass was a predictor of incident fractures within a 3-year follow-up, independent of the Fracture Risk Assessment Tool (FRAX) probability [38]. However, another large-scale observational study revealed that the co-occurrence of sarcopenia and osteoporosis instead of sarcopenia alone in both male and female American elderly individuals led to a significant risk of fracture during a ≥8-year follow-up [39]. The present study, which was the first population-based research with a follow-up of ≤13 years in Taiwan provides clinicians with additional strong evidence of the negative effect of sarcopenia on fracture risk.

This study had several limitations. First, despite using a national database, the sample size was small due to the perception of sarcopenia by clinicians in recent years. In addition, the diagnostic criteria for sarcopenia have been controversial among various expert groups but have gradually become accepted since 2010 [40]. Since our population-based data were collected from 2000 to 2013, the real number of nationwide patients with sarcopenia may have been underestimated. However, owing to the restriction by the policy for the data availability from the NHIRD and the limitations on the IRB approval for the data use, it is currently difficult to collect updated data from NHIRD for analysis. Future works on accessing the claims data up to date may be warranted to extend this research. Second, as previously mentioned, the diagnostic code for sarcopenia was developed only in the ICD-10-CM coding system in recent years. Although we utilized alternative ICD-9-CM code 728.2 to select the eligible patients possibly fulfilling the diagnosis of sarcopenia, failure to confirm the muscle mass and muscle strength, which were the main components for the diagnosis of sarcopenia [40], was the major weakness in this population-based study. Third, this study was a retrospective data analysis. Although evidence of the coexistence of osteoporosis may have predisposed the patients with sarcopenia to higher fracture risk [39], our study also failed to identify the patients’ bone mineral density or coexisting osteoporosis, leaving an unanswered question on the direct or indirect role of sarcopenia for increasing the fracture risk if osteoporosis was present. However, even with the abovementioned limitations, the strength of the present study was the first cohort design using national population-based data of patients in Taiwan with the longest follow-up period in the current literature, offering an important clinical evidence on the long term impact of sarcopenia on the risk of osteoporosis-related fracture.

## 5. Conclusions

The main findings of this retrospective, large-scale, cohort study indicated that patients with sarcopenia, especially females, had a significant risk of osteoporosis-related fracture within 13 years. Future studies are warranted to clarify sex differences in the fracture risk and to investigate the effectiveness of interventions to reverse the susceptibility to fracture in patients with sarcopenia.

## Figures and Tables

**Figure 1 jpm-12-00791-f001:**
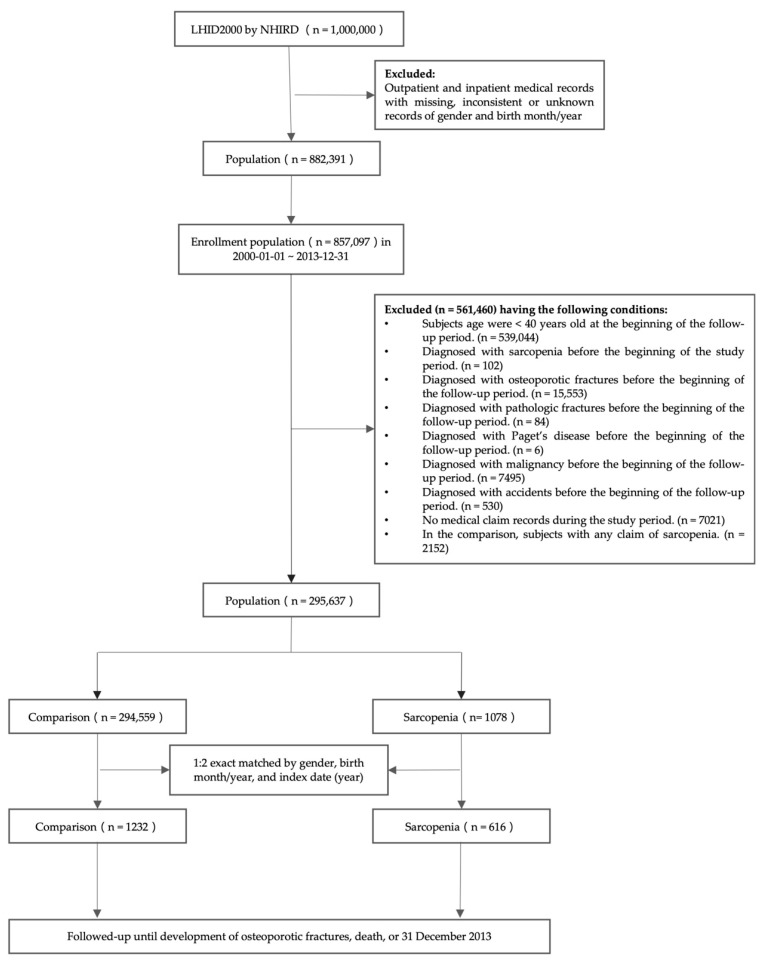
Flow chart of the study population selection.

**Figure 2 jpm-12-00791-f002:**
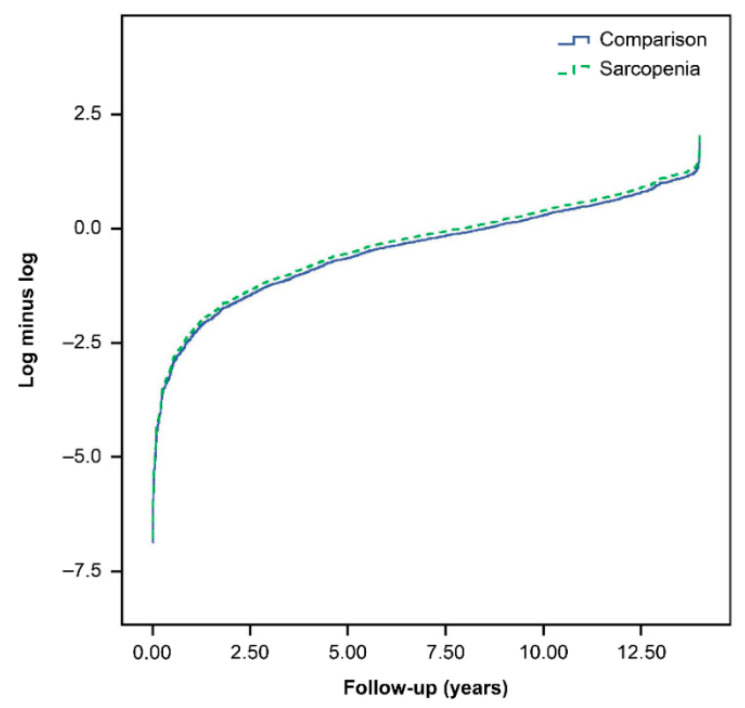
Plots of the log [−log (survival function)] versus the log of survival time for the comparison cohort and sarcopenia cohort.

**Figure 3 jpm-12-00791-f003:**
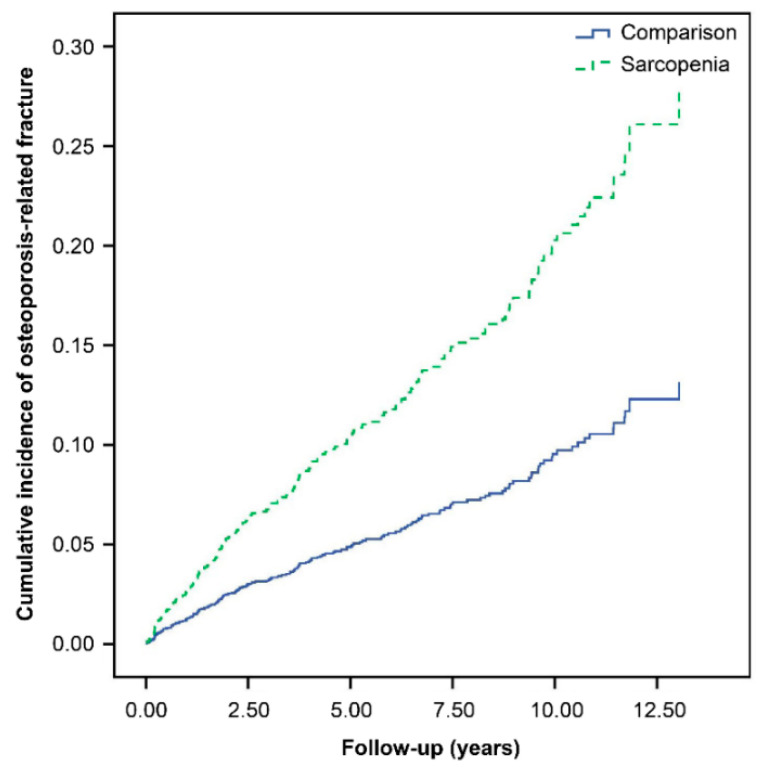
Cumulative incidence curves of osteoporosis-related fracture in individuals with and without sarcopenia.

**Table 1 jpm-12-00791-t001:** Clinical details of the sarcopenia and comparison cohorts.

Characteristics	*n* (%)		*p*-Value
	Comparison Cohort (*n* = 1232)	Sarcopenia Cohort (*n* = 616)	
Age, years			0.950
Mean ± SD	60.37 ± 12.00	60.41 ± 12.00	
Sex			Matched
Male	622 (50.49)	311 (50.49)	
Urbanization level ^a^			<0.001
1 (highest)	605 (49.11)	334 (54.22)	
2	370 (30.03)	146 (23.70)	
3	90 (7.31)	25 (4.06)	
4 (lowest)	14 (1.14)	3 (0.49)	
Unknown	153 (12.42)	108 (17.53)	
Insurance amount ^b^, NT$			0.498
Financially dependent	13 (1.06)	7 (1.14)	
1–19,999	567 (46.02)	262 (42.53)	
20,000–39,999	400 (32.47)	202 (32.79)	
≥40,000	152 (12.34)	92 (14.94)	
Unknown	100 (8.12)	53 (8.60)	
Confounding factors ^c^			
Rheumatoid arthritis	12 (0.97)	26 (4.22)	<0.001
Smoking habit	78 (6.33)	98 (15.91)	0.014
Alcohol use	26 (2.11)	26 (4.22)	0.700
Hypertension	327 (26.54)	197 (31.98)	0.012
Diabetes	253 (20.54)	243 (39.45)	<0.001
Dyslipidemia	159 (12.91)	265 (43.02)	<0.001
Obesity	2 (0.16)	8 (1.30)	0.005
Coronary heart disease	199 (16.15)	235 (38.15)	<0.001
Stroke	184 (14.94)	234 (37.99)	<0.001
Chronic obstructive pulmonary disease	163 (13.23)	281 (45.62)	<0.001
Depression	29 (2.35)	23 (3.73)	0.102
Cognitive dysfunction	51 (4.14)	85 (13.80)	<0.001
Parkinson’s disease	26 (2.11)	50 (8.12)	<0.001
Use of glucocorticoids	174 (14.12)	267 (43.34)	<0.001

SD, standard deviation. Values are shown as means ± SD or number (percentage). ^a^ Urbanization level was defined at the beginning of the follow-up period. ^b^ Insurance amount was measured as the average value during the follow-up period. ^c^ Confounding factors were defined before the survival date.

**Table 2 jpm-12-00791-t002:** Osteoporosis-related fracture frequency and HRs for osteoporosis-related fracture in both cohorts.

Population ^#^	Study Group	Osteoporosis-Related Fracture	PY	Rate ^a^	Crude HR (95% CI)	Adjusted HR ^b^ (95% CI)
Total	Comparison (*n* = 1232)	111	7599	14.61	1 (reference)	1 (reference)
	Sarcopenia (*n* = 616)	65	3585	18.13	1.23 (0.90, 1.67)	2.11 (1.47, 3.04) ^‡^
Female	Comparison (*n* = 610)	64	3894	16.44	1 (reference)	1 (reference)
	Sarcopenia (*n* = 305)	45	1804	24.95	1.50 (1.02, 2.20) *	2.40 (1.51, 3.81) ^‡^
Male	Comparison (*n* = 622)	47	3705	12.69	1 (reference)	1 (reference)
	Sarcopenia (*n* = 311)	20	1781	11.23	0.88 (0.52, 1.48)	1.53 (0.83, 2.82)

CI, confidence interval; HR, hazard ratio; PY, person years; Rate, incidence rate. ^a^ per 1000 person years. ^b^ Cox regression models were corrected for age, sex, urbanization level, rheumatoid arthritis, smoking habit, hypertension, diabetes, dyslipidemia, obesity, coronary heart disease, stroke, chronic obstructive pulmonary disease, cognitive dysfunction, Parkinson’s disease, use of glucocorticoids. * *p* < 0.05, ^‡^
*p* < 0.001. ^#^ stratified by sex.

## Data Availability

The data that support the findings of this study are openly available in Mendeley Data at http://doi.org/10.17632/hprf33swxy.1 (accessed on 21 January 2022).

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
