# Peer review of "Enhanced Risk of Osteoporotic Fracture in Patients with Sarcopenia: A National Population-Based Study in Taiwan"

_jpm, 2022, doi:10.3390/jpm12050791_

Round 1

Reviewer 1 Report

GENERAL COMMENTS
This study assessed the incidence and consequence of osteoporosis-related fractures among patients with sarcopenia in Taiwan.

I think it's a good study published on JPM after revising a few points in the manuscript.

SPECIFIC COMMENTS

Methods

Please move the last paragraph of the Data source section (2.1) to the Data selection section (2.3).

Results and Discussion

Describe the analysis associated with comorbidities.

Author Response

Submission no: jpm-1693077

Submission title:  Enhanced risk of osteoporotic fracture in patients with sarcopenia: A national population-based study in Taiwan

Dear editors from Journal of Personalized Medicine

Thank you for giving us the opportunity to revise our submitted manuscript. We tried our best to reply all the reviewer’s comments point-to-point and edit the manuscript accordingly. We hoped that the revised version fulfills the requirements of your esteemed journal.

Please kindly express our sincere and grateful appreciation to the anonymous reviewers. At last, please inform us if additional revisions are needed.

Reviewer 1

Question 1:

Methods 

Please move the last paragraph of the Data source section (2.1) to the Data selection section (2.3).

Reply 1:

Thanks for reviewer’s suggestion. We have revied the manuscript

Question 2:

2. Results and Discussion

Describe the analysis associated with comorbidities.

Reply 2:

Thanks for reviewer’s valuable suggestion. We added more description and discussion in revised text.

Revised text:

3. Results  (line 255-257 )

“The matched-patient analysis between the sarcopenia and control cohorts revealed significant differences for the parameters of urbanization level and many of the comorbidities including rheumatoid arthritis, hypertension, diabetes, dyslipidemia, obesity, coronary heart disease, stroke, chronic obstructive pulmonary disease, cognitive dysfunction and Parkinson’s disease.

4. Discussion  (line 324-342 )

“Sarcopenia is a musculoskeletal syndrome referred to increasing age or secondary to disease [1]. However, several chronic diseases may be also directly or indirectly associated with sarcopenia. Evidence has disclosed that high prevalence of sarcopenia is found in patients with several age-related diseases including cardiovascular diseases, dementia, diabetes and respiratory disorder [2]. Several mechanism including decreased activity, sedentary behavior, inflammation and malnutrition might explain the correlation between sarcopenia and these age-related diseases. Decreased physical activity and sedentary life style, which are well-known risk factors for sarcopenia [3], are common in older population [4]and also highly prevalent in patients with chronic heart failure [5], diabetes [6] and respiratory diseases [7]. In addition, chronic inflammation is characteristic of aging process, deeply connected with age-related diseases and shown to prompt muscle wasting and pathological muscle loss [8]. On the other hand, older patients with cognitive disorders are susceptible to malnutrition [9], which is also a critical factor resulting in sarcopenia [10]. All the above-mentioned cofounding factors may therefore result in the sarcopenic patients concomitantly suffering more comorbidities, acting in concert with our finding that older patients with sarcopenia had higher prevalence of rheumatoid arthritis, hypertension, diabetes, dyslipidemia, obesity, coronary heart disease, stroke, chronic obstructive pulmonary disease, cognitive dysfunction and Parkinson’s disease than those without sarcopenia. “

Reviewer 2 Report

1.  Period of study is of great concern (2000-2013).  Data is about a decade old. So this study must be extended to present day or atleast upto 2021. 

2.  Authors have mentioned that this study have lot of limitations. With these limitations this paper cannot be published.  So line nos. 285 - 302 must be rewritten highlighting the positives of the study.

3.  Only if points (1) and (2) are met this paper can be published.

4.  Instead of using "WE" the authors may use "THIS STUDY" OR RELATED PHRASE.

5.  In line nos. 309 - 320, expansions must be included.

6. If plagiarism is not checked for this paper then it must be carried out and it should be less than 20%
